# The Burden of Congenital Hypothyroidism Without Newborn Screening: Clinical and Cognitive Findings from a Multicenter Study in Algeria

**DOI:** 10.3390/ijns11030078

**Published:** 2025-09-15

**Authors:** Adel Djermane, Yasmine Ouarezki, Kamelia Boulesnane, Sakina Kherra, Fadila Bouferoua, Mimouna Bessahraoui, Nihad Selim, Larbi Djahlat, Kahina Mohammedi, Karim Bouziane Nedjadi, Hakima Abes, Meriem Bensalah, Dyaeddine Lograb, Foued Abdelaziz, Dalila Douiri, Soumia Djebari, Mohamed Seghir Demdoum, Nadira Rouabeh, Meriem Oussalah, Guy Van Vliet, Asmahane Ladjouze

**Affiliations:** 1Faculty of Medicine, University of Health Science, Algiers 16028, Algeria; a.djermane@univ-alger.dz (A.D.);; 2Department of Paediatrics, Hassan Badi Public Hospital, Algiers 16051, Algeria; 3Department of Paediatrics, Bab El Oued Teaching Hospital, Algiers 16008, Algeria; 4Department of Paediatrics, Nafissa Hamoud Teaching Hospital, Algiers 16040, Algeria; 5Department of Paediatrics, Beni Messous Teaching Hospital, Algiers 16026, Algeria; 6Department of Paediatrics, Canastel Children’s Hospital, Oran 31130, Algeria; 7Faculty of Medicine, University of Oran, Oran 31000, Algeria; 8Faculty of Medicine, University of Annaba, Annaba 23000, Algeria; 9Department of Paediatrics, Saint Therese Clinic, Annaba 23000, Algeria; 10Independent Researcher, Mascara 29006, Algeria; 11Department of Pediatrics, Ain Taya Public Hospital, Algiers 16019, Algeria; 12Department of Paediatrics, Oran Teaching Hospital, Oran 31000, Algeria; 13Department of Pediatrics, Douira Teaching Hospital, Algiers 16049, Algeria; 14Faculty of Medicine, University of Blida, Blida 09000, Algeria; 15Endocrinology Unit, Central Hospital of Army, Algiers 16208, Algeria; 16Independent Researcher, Boussada 28200, Algeria; 17Independent Researcher, Annaba 23000, Algeria; 18Department of Paediatrics, Bologhine Public Hospital, Algiers 16030, Algeria; 19Independent Researcher, El-Oued 39002, Algeria; 20Department of Paediatrics, Setif Teaching Hospital, Setif 19000, Algeria; 21Faculty of Medicine, Setif University, Setif 19137, Algeria; 22Department of Pediatrics, University of Montréal, Sainte-Justine Hospital, Montreal, QC H3T 1C5, Canada

**Keywords:** congenital hypothyroidism, newborn screening, neurodevelopmental, IQ, treatment, Algeria

## Abstract

The absence of biochemical newborn screening (NBS) delays the diagnosis and treatment of congenital hypothyroidism (CH), resulting in irreversible neurodevelopmental damage. To determine the age at diagnosis for CH among Algerian children and to describe its clinical and biological characteristics, etiology, and outcome, we conducted a multicenter retrospective cohort study involving 288 children with CH across 20 pediatric centers between 2005 and 2023. The median age at diagnosis was 1.6 months, and only 28% of patients started treatment before 30 days. Prolonged neonatal jaundice was the most frequently presented symptom (58%), severe CH (fT_4_ < 5 pmol/L) was observed in 35% and 52% received an insufficient initial dose of L-T_4_. The median IQ of the 47 patients tested was 86; 11% had an IQ < 70, and a negative correlation was found between age at diagnosis and IQ (r = −0.48, *p* = 0.001). In children reassessed at age 3, 51% had normal thyroid function, indicating transient CH. Delayed diagnosis and suboptimal treatment of CH remain major challenges in Algeria, leading to substantial neurodevelopmental deficits. Pediatricians must remain cognizant of early clinical signs of CH to allow for timely diagnosis and intervention. Biochemical NBS for CH in Algeria is needed.

## 1. Introduction

Congenital hypothyroidism (CH) is one of the most common causes of preventable intellectual disability in children. In countries where universal newborn screening (NBS) programs are established, CH is routinely detected within the first two weeks of life—allowing for early initiation of levothyroxine therapy and preventing long-term neurodevelopmental impairments [1]. However, in the absence of screening, diagnosis often relies on clinical suspicion alone, which may be delayed due to the subtle or nonspecific nature of early symptoms [2].

In Algeria, there is currently no nationwide NBS program for CH. As a result, the diagnosis is frequently based on the recognition of clinical signs, such as prolonged neonatal jaundice, constipation, hypotonia, or delayed growth—signs that often appear after the critical period for optimal neurological development. This delayed recognition contributes to late initiation of treatment and increases the risk of permanent psychomotor and intellectual deficits [3].

In this context, our study aims to evaluate, for the first time on a national scale, the age at diagnosis of CH in Algeria. It also seeks to characterize the clinical and biological features at presentation, identify etiological patterns, assess the adequacy of treatment, and investigate the neurodevelopmental consequences of delayed diagnosis. Our ultimate goal is to provide evidence to support the implementation of a universal NBS program and to raise awareness among healthcare professionals and decision makers about the importance of the early detection of CH.

## 2. Methods

This was a multicenter observational cohort study.

### 2.1. Objectives

The primary outcome was to determine the age at diagnosis of patients with CH in Algeria, where newborn screening is not established.

The secondary outcomes were to determine the clinical and biological characteristics at presentation, the etiology of CH, and the neurodevelopmental outcome, as well as to explore the correlation between neurodevelopmental outcomes and age at diagnosis.

### 2.2. Study Design and Population

We retrospectively reviewed the medical records of 432 children followed for CH diagnosed between February 2005 and September 2023 across twenty pediatric departments in Algeria. Neurodevelopmental assessments were conducted prospectively in a subset of patients as part of standardized follow-up evaluations.

The study population included all children with CH aged 0 and 18 years who attended an outpatient clinic from January 2017 to December 2023.

The exclusion criteria were central hypothyroidism, Down syndrome, patients who were not treated, and those who experienced early discontinuation of treatment or were lost to follow-up.

### 2.3. Data Collection

The following data were collected for analysis: reason for referral, clinical characteristics at diagnosis, auxological data, serum TSH and fT_4_, radiological evaluation (knee X-rays), etiology based on imaging, and dose of L-T4 treatment. TSH values above 100 mU/L (not reported precisely) were included in the analysis by assigning them a value of 100 mU/L. Treatment under the recommended dose was considered inadequate [4,5].

### 2.4. Definitions

Primary hypothyroidism was defined by elevated serum TSH (age-adjusted, minimal TSH > 8 mU/L) and/or a low fT_4_ (<10 pmol/L). Due to insufficient follow-up data, subclinical hypothyroidism could not be reliably identified or analyzed.

Two etiological groups of CH were defined based on the results of thyroid ultrasound (US) and/or pertechnetate scintigraphy: thyroid dysgenesis (TD), including (athyreosis, ectopic gland, and orthotopic hypoplasia) and gland in situ (GIS) of a normal or increased size (including all patients with normal scintigraphy/US and patients with a goiter detected clinically). Cases with GIS were considered suggestive of dyshormonogenesis (DH).

The initial L-T_4_ dose was considered insufficient when it was below 9 µg/kg/day in the first 3 months, and below 6 µg/kg/day after 3 months.

### 2.5. Neurodevelopmental Assessment

The neurodevelopmental status of the children was determined using several measures, including psychomotor delay (defined as a delay in attaining developmental milestones) and school progression (absence of schooling, grade repetition, or poor school performance) for the school-aged children (defined as 6–18 years). In addition, IQ was measured in some patients using the Arabic version of the Wechsler Intelligence Scale for Children—IV for children aged 6 to 16 years; children under 6 years of age were assessed using age-appropriate standardized tools such as the Wechsler Preschool and Primary Scale of Intelligence (WPPSI-III), the Columbia Mental Maturity Scale, and the Kohs Block Design Test [6,7]. Scores of IQ < 70 are considered to indicate intellectual disability, and ≥85 are considered normal intelligence. We analyzed and compared the characteristics between low IQ (<85) and normal IQ (≥85) groups.

### 2.6. Reassessment

In some children who did not undergo scintigraphy before treatment, treatment was stopped after 3 years, and serum TSH and fT_4_ were reassessed. CH was then classified as either transient if TSH did not rise after treatment withdrawal, and as permanent otherwise [8].

### 2.7. Ethics

An ethics statement is not applicable because this study is based exclusively on a clinical audit.

### 2.8. Statistical Analysis

The data were expressed as mean ± SD or median (range) as appropriate. For comparisons between two groups, independent-samples *t*-tests, Mann–Whitney or Chi-square (χ^2^) tests were used. For more than two groups, one-way analysis of variance (ANOVA) was used. Correlational analyses were performed using Pearson’s correlation coefficients. To identify risk factors for low IQ and psychomotor delay, multivariate analyses such as linear regression, logistic regression, and multiple regression models were applied. Receiver Operating Characteristic (ROC) analysis was used to evaluate the performance of a diagnostic test. Epi Info 7, Excel, and Medcalc were used to collect and analyze the data.

## 3. Results

Of 432 CH patients, 288 (66.6%) were included in the study, with 144 excluded for reasons such as missing data, central hypothyroidism, Down syndrome, or lack of data on treatment/follow-up. Figure 1 provides a visual representation of the patient selection and exclusion process, leading to the final group of 288 patients whose clinical characteristics were analyzed in the study.

### 3.1. Characteristics of the CH Population

*Age at diagnosis*: The median age at diagnosis was 1.6 (0.05–150) months, but the median age at the initiation of treatment was 2 months, ranging between <1 and 156 months. Less than half of the patients (35%) were diagnosed before 1 month, while more than 37% of the patients were diagnosed after three months, and 18% after 1 year of age (Figure 2). The oldest CH case was diagnosed at 150 months, presenting with short stature and developmental delay.

*Sex ratio*: There were 159 females (55%) and 129 males (45%), leading to a female-to-male ratio of 1.23.

*Consanguinity and family history*: Consanguinity was reported in 75 patients (26%), a family history of thyroid dysfunction was noted in 105 patients (36%), and CH was reported in 27 (9%) siblings from 12 families. Moreover, an extrathyroidal congenital abnormality was present in 44 patients, with cardiac defects being the most common (8%) (Table 1).

*Birth characteristics*: The average gestational age was 38 weeks, and prematurity was noted in 27 patients (9%), while 11% of the patients were born SGA. The mean birth weight, length, and head circumference were 3.2 kg, 49.6 cm, and 34.4 cm, respectively (Appendix A).

*Cause of referral and signs at presentation* (Figure 3): The most common symptom leading to the suspicion of CH and referral was jaundice for more than 10 days (36.5%), whereas it was clinically observed at diagnosis in 58% (Appendix A). Other symptoms included constipation, psychomotor delay, short stature, hypotonia, and goiter. Only 13% had a targeted screening with serum TSH during the neonatal period because of a family history of hypothyroidism (sibling with CH or mother with hypothyroidism).

Forty-two (15%) patients had short stature (Height < −2 SD) at diagnosis, and seventeen (6%) had a BMI > +2 SD. The reason for referral had a significant impact on the age at diagnosis: jaundice and constipation were common before 3 months, while psychomotor delay and short stature were reported in patients diagnosed after 12 months (Appendix A). Those who presented with prolonged jaundice at a median age of one month (range: 0.05–34) or were screened at a median age of 0.3 months (range: 0.07–12) had the youngest age at diagnosis, whereas those who presented with short stature had the oldest age at diagnosis (median 39 months (range: 2.33–150) (Figure 4).

*Biological data* (Table 2): The diagnosis of hypothyroidism was confirmed through the measurement of serum TSH and fT_4_ prior to treatment in all patients. The median TSH value before treatment was 65.35 mU/L. Mean ± SD fT4 levels were 7.6 ± 6.2 pmol/L (range 0.01–20). A total of 43% were found to have extremely high TSH levels (>100 mU/L), 56% had low fT_4_ levels (<10 pmol/L), and 34% had fT_4_ levels <5 pmol/L, indicating severe CH.

### 3.2. Imaging Exams and Aetiological Groups

Knee radiography was performed in 97 patients (34%): no ossification center was shown in 34 full-term infants, indicating a prenatal onset of hypothyroidism. Thyroid US was performed in 251 patients (87%), and pertechnetate scintigraphy in 137 (48%). The GIS group was the largest etiological group (150 cases, 52%), with goiters accounting for only 9% of the cases, while thyroid dysgenesis was found in 109 cases (38%). In 29 (10%) cases, the etiological group could not be defined because of the lack of an imaging study. The dysgenesis group showed significantly more severe forms, while consanguinity and female sex were more frequent in the GIS group (Table 3).

### 3.3. Treatment

Levothyroxine (L-T_4_) treatment was initiated at a median age of 2 months, with a range between 0.06 (2 days) and 150 months. Only 82 (28%) of the patients were started on L-T_4_ before 30 days of age. Among the 288 patients included, 228 (79%) began L-T4 treatment within 15 days following the biological diagnosis of hypothyroidism, while 42 (15%) patients began treatment more than one month after diagnosis. The mean starting dose of L-T_4_ was 6.9 ± 4.1 (range 0.6–25.6 µg/kg/day). Patients aged less than 1 month at diagnosis received a higher dose (7.8 ± 3.7 µg/kg/day) (Appendix A). One hundred forty-nine (52%) received an insufficient initial L-T_4_ dose, with 51% receiving less than 9 µg/kg/day in the first 3 months and 53% receiving less than 6 µg/kg/day after 3 months (Figure 5).

### 3.4. Outcomes

#### 3.4.1. Transient vs. Permanent CH

Forty-seven patients (16%) underwent biological reassessment after stopping treatment, of which twenty-four (51%) had normal thyroid function, indicating transient hypothyroidism (Table 4).

In patients with transient CH, initial TSH levels were significantly lower and fT_4_ was significantly higher at diagnosis than in those with permanent CH. The transient CH cases were more frequent in the GIS group (Table 4).

#### 3.4.2. Neurodevelopmental Assessment (Table 5)

A total of 16 patients (6%) were reported to have a language delay, 5 (2%) had hearing loss, while 43 (15%) were described as having a psychomotor delay.

School progression data were available for 88 of 206 school-aged children (43%): 27 (31%) had repeated a grade or had left regular school.

A high rate of delayed treatment beyond the first month of life was observed among patients with neurodevelopmental impairments. This was particularly pronounced in extreme cases, including psychomotor delay (97.7%), language delay (87.5%), school failure (77.8%), and IQ below 85 (>85%).

Among children diagnosed at age 12 months or older, 6/52 (11.5%) were found to have a severe developmental delay. The ROC plot analysis showed that the threshold line for an age of 1.94 months is associated with psychomotor delay with a sensitivity of 87.2%, a specificity of 66.7%, and an AUC of 0.815, *p* < 0.001 (Figure 6). Among the one hundred and fifty (151) children older than 1.94 months at diagnosis, sixty (40%) were labeled with psychomotor delay, nine (6%) of whom had severe CH. Using multiple regression, children with CH diagnosed after 2 months of age had an OR of 7.77 [2.88–20.67] of having psychological delay, while this OR was 0.29 [0.07–1.18] in those diagnosed before 1 month of age (Appendix A).

#### 3.4.3. IQ Evaluation

Out of 47 (16%) patients with an IQ evaluation (Table 5), only 62% had a normal value, while 11% had an IQ < 70, the WHO definition of intellectual disability (Figure 7). The median age at IQ assessment was 5 years (range: 2.9–9). The mean IQ was 86.1 ± 15.2 (50–112). This result was significantly lower than the theoretical mean IQ of 100 in the general population (*p* < 0.0001) (Figure 7). Among the children who underwent IQ testing, only three were later classified as having transient congenital hypothyroidism, and all were within the normal range (93, 100, and 109). Thirty-seven of the forty-seven IQ assessments (79%) were performed at the same center with standardized methods, minimizing inter-center variability and improving data consistency.

Children with IQ < 85 (*n* = 18) did not differ from those with IQ ≥ 85 (*n* = 29) in terms of gender, age at IQ test, and etiology (Table 6).

At the start of treatment, children with IQ < 85 were significantly older (median 3.75 months, range: 0.2–69) than children with IQ ≥ 85 (median 1.1 months, range: 0.1–42.7). The IQ ≥ 85 group had started treatment before one month (45%), and with a dose higher than 9 µg/kg/day than those with IQ < 85 (38% vs. 22%). Seven (15%) patients with a normal IQ have a cognitive disharmony (Table 6).

Our analysis shows a significant negative correlation between IQ and age at diagnosis (r = −0.48, *p* = 0.001) (Figure 8).

Logistic regression indicates that dysgenesis significantly increases the risk of low IQ (OR = 5.55 [1.1–25.7]). A second IQ test on 11 CH patients at a median age of 8 years (range: 4.5–10) showed improvement to an IQ ≥ 85 in five cases (45.5%).

## 4. Discussion

In the absence of a standard biochemical NBS program, the diagnosis of CH relies on a systematic and rigorous clinical examination of newborns at birth and during the first weeks of life. Despite this, not all patients are diagnosed in time to avoid intellectual disability. Even though the inclusion period spans more than 15 years, most pediatric endocrinology centers participating in this study have existed for less than 10 years; so, despite an estimated 10-year recruitment period, the recruitment concerned only some hospital-based pediatric clinics, and therefore does not allow for an estimation of the population prevalence of CH. However, published studies indicate that the incidence of CH ranges from 1 in 7000 to 1 in 10,000 births in populations without screening, while this rate increases to between 1 in 1100 and 1 in 3000 births when screening is implemented, depending on the TSH threshold used [9,10,11,12].

*Clinical Diagnostic Challenges*: The difficulty in early clinical diagnosis stems from frequently subtle or absent symptoms at birth [9,13]. In our study, jaundice emerged as the primary diagnostic clue, with a median age at presentation of 1.66 months. This aligns with established knowledge that CH often manifests as prolonged neonatal jaundice [2,13,14] accompanied by other characteristic symptoms such as constipation, macroglossia, delayed fontanel closure, umbilical hernia, and coarse facial features [14,15,16]. The clinical diagnosis remains particularly challenging, as most affected infants exhibit highly nonspecific symptoms, with only about 5% of CH cases presenting sufficiently distinctive signs within the first few days of life to allow for a prompt diagnosis [17]. Furthermore, it is concerning that approximately 5–10% of newborns with CH are not detected by primary screening programs, irrespective of whether the screening targets thyroxine (T_4_) or thyroid-stimulating hormone (TSH) [18].

*Delayed diagnosis of CH* remains alarmingly common in many parts of the world, when NBS programs are inadequate or absent, showing persistent gaps in early detection [19]. In regions lacking widespread NBS, the median age at diagnosis is significantly delayed, often exceeding 45 days, which contributes to poorer developmental outcomes (Table 7) [3,15,16,20,21,22,23,24,25,26,27,28,29]. Aligning with global data, our results demonstrate a concerning delay in CH diagnosis, with a mean age at diagnosis of 2 months, and 65% of cases identified beyond the neonatal period (>1 month).

### 4.1. The Consequences of Delayed Diagnosis and Treatment

The effects of delayed diagnosis and treatment of CH are often severe and irreversible, significantly impacting both development and quality of life [15]. Early studies indicate that delaying treatment beyond three months of age, and even more beyond the age of six months, causes irreversible intellectual disability with a higher risk of lower IQ scores and neuropsychological deficits, collectively referred to as cretinism [18,30,31,32]. Late initiation of L-T_4_ therapy has been shown to negatively affect cognitive functioning and overall well-being [33]. Research by Leger et al. provides strong evidence that, during early adolescence, there is a clear link between disease severity at the time of diagnosis, the adequacy of treatment during follow-up, and poor school performance among individuals treated for CH from the neonatal stage [34,35]. Several factors have been identified as predictors of worse intellectual outcomes in children with CH, including initial serum T_4_ levels at diagnosis, the timing of treatment initiation, prolonged time to normalize thyroid hormone levels, maternal education, socioeconomic status, and the frequency of clinic visits during the first year of life [5,31,32,36,37]. Cognitive problems may persist despite treatment, including difficulties with visual–spatial abilities, language development, fine motor skills, memory, and attention [38]. Beyond the cognitive domain, untreated cases often exhibit severe growth impairments, including stunted growth and delayed bone maturation [39]. Neurological complications such as spasticity, gait abnormalities, dysarthria, mutism, and behavioral disorders may also develop in affected individuals when treatment is delayed [33].

### 4.2. Etiological Diagnosis and Genetic Factors in Congenital Hypothyroidism

CH is predominantly accounted for by two main etiological categories: thyroid dysgenesis (TD) and dyshormonogenesis (DH). TD represents approximately 80–85% of CH cases and includes conditions such as ectopy, athyreosis, and orthopic hypoplasia, which are rarely linked to mutations in transcription factor genes like *TSHR*, *PAX8*, and *NKX2-1* [8,40]. Less commonly, DH results from defects in thyroid hormone synthesis, predominantly associated with mutations in the *DUOX2*, *TPO*, and *TG* genes [41,42,43].

Our study found a high proportion of GIS cases (43%), indicating a possible increased prevalence of CH due to DH. As goiter was not consistently present, in the absence of perchlorate testing and genetic analysis, a definitive diagnosis of DH could not be made.

Populations in which consanguineous marriages are common have reported a rising incidence of CH and DH [44]. Such marriages significantly increase the likelihood of inheriting recessively transmitted genetic mutations, contributing to a higher rate of CH related to mutations in genes like *DUOX2* or *TPO* compared to populations with lower rates of consanguinity [45]. Arab countries with high consanguinity rates also show a relatively high prevalence of CH based on newborn screening programs. For example, CH incidence ranges from 1/1778 in the UAE to 1/2939 in Saudi Arabia [13,28,46]. Furthermore, consanguinity is associated with an increased incidence of congenital abnormalities, including umbilical hernia, congenital heart disease, genitourinary malformations, and cleft palate [8,39,47,48,49]. However, these malformations are not typically linked to autosomal recessive mutations commonly associated with consanguinity, such as *DUOX2* and *TPO* genes, which are typically linked to isolated thyroid dysfunction. In contrast, autosomal dominant mutations in genes like *PAX8*, *NKX2-1*, and *NKX2-5* are more often associated with syndromic forms or extrathyroidal anomalies [50].

In Algeria, consanguinity is reported in nearly 30% of marriages [51], which may explain the high proportion of gland-in situ cases and of congenital abnormalities.

### 4.3. Permanent and Transient Forms of CH

The re-evaluation at three years of age revealed that 51% of children with CH tested displayed normal thyroid function, suggesting a substantial number of cases had transient CH. This implies that the global prevalence of CH could include patients with temporary forms of the condition. Previous studies have indeed reported that up to 60% of children on L-T_4_ replacement therapy experienced transient hypothyroidism leading to the discontinuation of L-T_4_ treatment [52,53]. Approximately 17% to 40% of children diagnosed with CH by NBS programs were later found to have transient hypothyroidism [54]. Several factors may contribute to this finding, including variations in assay threshold, maternal TSH receptor-blocking antibodies, maternal antithyroid drugs, genetic defects such as *DUOX2* mutations, and potential iodine imbalance in the population. However, the underlying mechanism often remains unknown [54,55]. Permanent CH is likely in the presence of TD and of an initial TSH > 100 mU/L [56]. In the absence of these, the need for continuation of L-T_4_ replacement therapy should always be reassessed in children being treated for CH at three years of age [8,57].

### 4.4. Treatment of CH

When CH is diagnosed late, the primary focus is on prompt initiation of treatment and addressing existing complications [58]. The standard treatment for CH involves L-T_4_ replacement therapy, typically initiated at a dose of 10–15 mcg/kg/day [57,59,60]. However, findings from our study indicate that 52% of patients received an inadequate initial L-T_4_ dose and 15% initiated treatment more than one month after diagnosis. This high proportion of delays and suboptimal initial L-T4 dosing likely reflects a combination of factors, including variability in physician practice, hesitancy to initiate higher doses in very young infants, late referrals, limited access to care, and, in some cases, limited adherence to evolving guidelines at the time of diagnosis. It is clear that the treatment with very low doses may delay thyroid hormone normalization and increase the risk of associated complications [59]. The aim of the treatment is to normalize thyroid function and maintain fT_4_ levels in the upper half of the age-specific reference range during the first three years of life [61]. Even when the diagnosis of CH is delayed, immediate treatment initiation is critical to prevent further deterioration and potentially improve existing symptoms. Some recent studies suggest that an individualized dosing based on the etiology and severity of CH can optimize outcome [62], with children with thyroid dysgenesis generally requiring higher L-T_4_ doses than children with DH [4,19].

Regular thyroid function testing (TSH and fT_4_) is essential to ensure treatment adequacy and to prevent complications from overtreatment or undertreatment. Overtreatment can lead to adverse effects, including hyperthyroidism, while undertreatment may result in persistent hypothyroidism and increase the likelihood of developmental delay [4,63].

### 4.5. Neurodevelopmental Data

*IQ Findings*: The limited number of IQ assessments, 47(16%), was primarily due to the lack of availability of standardized cognitive testing in the majority of participating centers, rather than selective clinical indication. This shows the difficulties in measuring cognitive outcomes in the larger CH population and is a potential selection bias. This highlights the need for more research on how early diagnosis and treatment affect cognitive development. Our study found that children with CH had a mean IQ of 87.1 ± 15.2, which is lower than the general population norm (100 ± 15). Additionally, IQs below 70 (11%) were significantly more common among CH patients compared to the general population [9]. In contrast, some follow-up studies suggest that the global IQ in CH children treated early thanks to NBS does not differ from that of controls [64,65,66].

#### 4.5.1. Educational Outcomes and Cognitive Impairments

In school-aged children, 31% repeated a grade or failed school, indicating notable academic challenges. However, children who received early treatment generally performed within the normal range. Nevertheless, generalized learning difficulties were still found in 20% of CH children [67]. Other studies report mild cognitive impairments, including lower mean IQ scores and subtle deficits in attention, memory, fine motor skills, and quality of life [68,69,70,71,72,73].

A higher initial dose of levothyroxine combined with very early treatment initiation may lead to better cognitive outcomes [74]. A few patients with severe CH may still have subtle cognitive and motor deficits, and lower educational attainment despite early treatment with a high initial L-T_4_ dose [70,74,75,76]. Moreover, the long-term neurodevelopmental outcomes in patients with CH appear to be associated with the severity of hypothyroidism and the subsequent rapid normalization of TSH [8]. In our study, euthyroidism was achieved in all patients at the time of neurocognitive assessment, and mean scores of both developmental quotient and intelligence quotient were lower than the general population, with differences in primary school performance. Even in CH patients screened at birth, mild non-verbal learning disabilities, and less than satisfactory scores for educational attainment, behavior, and motor skills were reported in children with severe CH [76]. However, the finding of CH patients with subnormal IQ suggests that neurodevelopmental rescue should not be taken for granted even in the era of neonatal screening [64,77]. Children diagnosed and treated after 3 months of age are at high risk of permanent cognitive impairment [18,33]. In our study, a diagnosis after 2 months appears to represent the critical threshold in our ROC analysis, with an OR of 7.3 for developing an intellectual disability. In an Indonesian study, 72% of patients with CH (median age 9 years) had a full-scale IQ score <70 (classified as intellectual disability), with late initiation of treatment specifically correlating with reduced performance IQ [33]. Severe cases (TSH >30 mU/L) are 5–14 times more likely to exhibit developmental delay in cognition and language [78,79]. Despite these challenges, some follow-up studies demonstrate a positive trajectory in cognitive functioning among many patients with CH, particularly those diagnosed and treated early. In our study, five patients showed improvement in their IQ scores over time. This highlights the importance of early intervention and of consistent follow-up to maximize developmental outcomes [1].

#### 4.5.2. Preventing Delayed Diagnosis and Reducing Cognitive Risks in Congenital Hypothyroidism

Preventing delayed diagnosis and reducing cognitive risks related to CH requires a comprehensive approach involving healthcare systems, provider education, and public awareness [61]. An important advancement in preventive medicine has been the implementation of NBS for CH, which has significantly changed the natural history of this condition [33,80,81]. NBS has been crucial in lowering the incidence of intellectual disability associated with untreated CH [38,82]. Based on available data, it is estimated that approximately 25% of children born with clinically diagnosed CH may have experienced overt disability before the adoption of NBS [12,83]. Early detection through NBS is associated with better neurocognitive outcomes in CH [19,61,83].

#### 4.5.3. Challenges in Timely Screening Implementation

Despite its success, NBS programs face some challenges, including early hospital discharges, which can complicate the timing for blood sample collection. In resource-limited settings, efforts should focus on targeted screening approaches for high-risk infants or the development of more affordable screening methods [10]. Additionally, international organizations can assist in establishing and supporting NBS programs in underserved regions, thereby improving early identification and treatment rates globally [84].

#### 4.5.4. Education and Public Awareness Initiatives

##### Healthcare Provider Education

Ongoing education for healthcare providers is essential to enhance their ability to recognize, diagnose, and manage CH early in the absence of a national NBS program. This is particularly relevant for primary-care providers, who are often the first to encounter subtle signs of CH in infants who were not screened at birth. Regular training programs should focus on identifying subclinical presentations, interpreting screening results, and understanding updated treatment protocols.

##### Public Awareness Campaigns

Educational campaigns targeting parents and caregivers are also vital. These efforts should emphasize the importance of NBS and raise awareness about the signs of thyroid dysfunction in infants and children, such as delayed growth, feeding difficulties, or developmental delay. Reaching underserved communities—where healthcare access is often limited—requires tailored communication strategies to ensure equitable access to information.

## 5. Limitations

The study has several limitations. First, its retrospective design introduces potential biases, including incomplete data and reliance on clinical records. Second, the non-exhaustive recruitment of patients may limit the generalizability of the findings. Third, the small number of IQ evaluations (*n* = 47) restricts the ability to draw definitive conclusions about neurodevelopmental outcomes. Future prospective studies with larger sample sizes and standardized neurodevelopmental assessments are needed to address these limitations.

## 6. Conclusions

This study highlights the critical challenges in managing CH in Algeria, including delayed diagnosis, suboptimal treatment practices, and the impact of consanguinity on disease etiology. The findings underscore the urgent need for universal NBS to facilitate early diagnosis and treatment, as well as adherence to international guidelines for L-T_4_ dosing. Public health interventions to reduce consanguinity and improve access to genetic counseling should also be prioritized. Preventing delayed diagnosis of CH and minimizing its cognitive impact depend on a robust infrastructure for NBS, adequate provider training, and community education initiatives. While NBS has significantly reduced the burden of intellectual disability associated with CH, challenges remain, particularly in resource-limited regions. Finally, further research is needed to evaluate the impact of these interventions on neurodevelopmental outcomes in Algerian children with CH.

## Figures and Tables

**Figure 1 IJNS-11-00078-f001:**
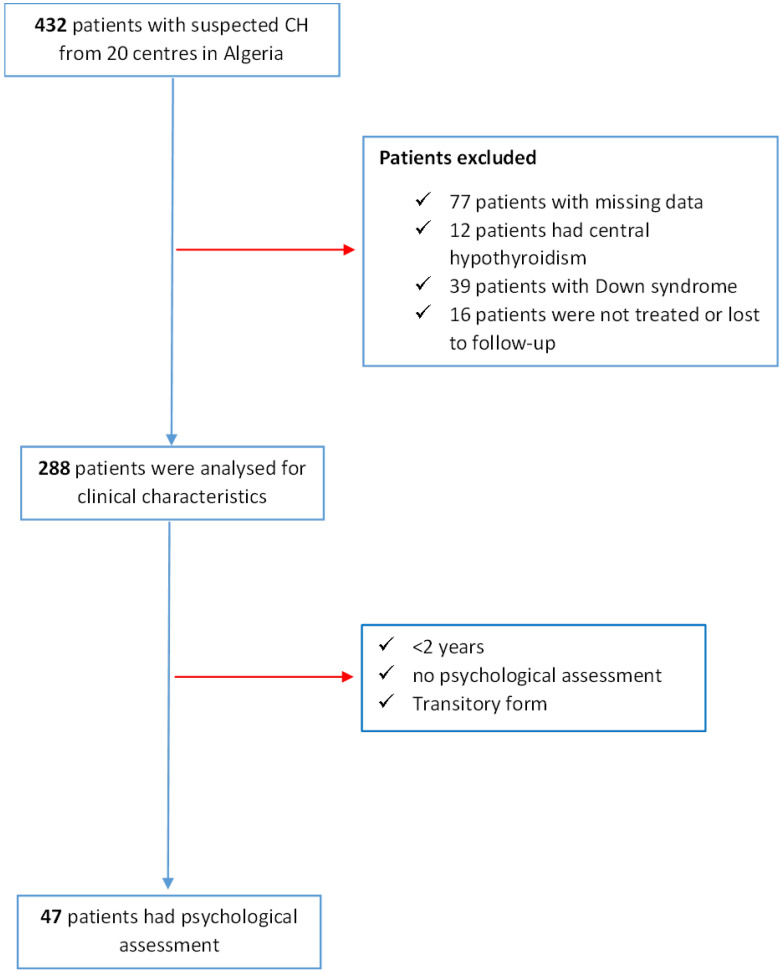
Flowchart of the study.

**Figure 2 IJNS-11-00078-f002:**
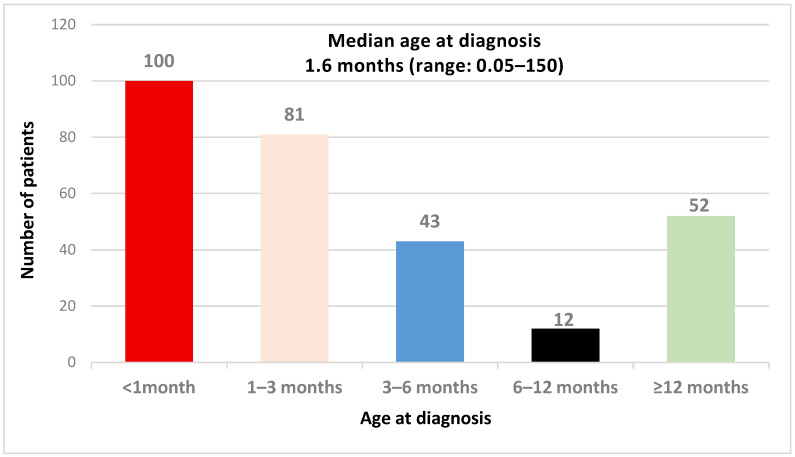
Age at diagnosis in months.

**Figure 3 IJNS-11-00078-f003:**
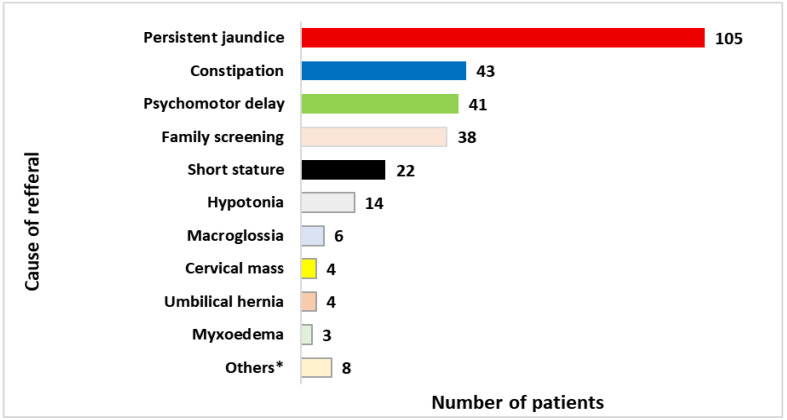
Cause of referral. * No CH-associated clinical signs or not reported.

**Figure 4 IJNS-11-00078-f004:**
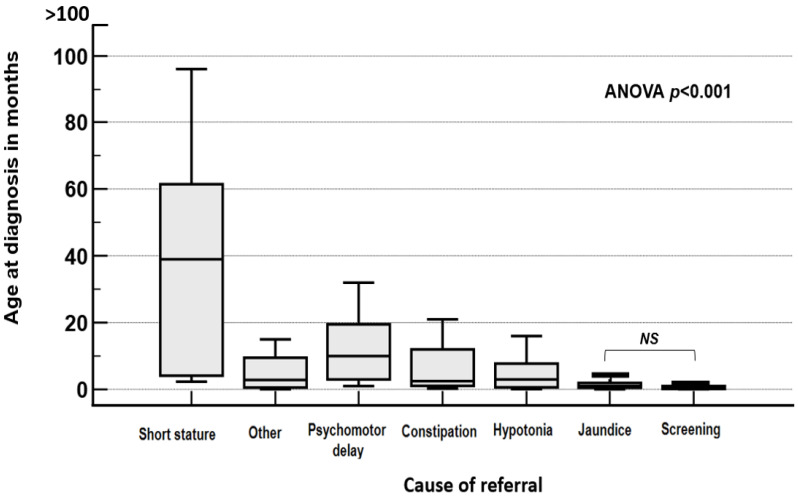
Age at diagnosis in months according to the cause of referral. NS: No significant difference. Boxplot: middle line = median, box = interquartile range, whiskers = range.

**Figure 5 IJNS-11-00078-f005:**
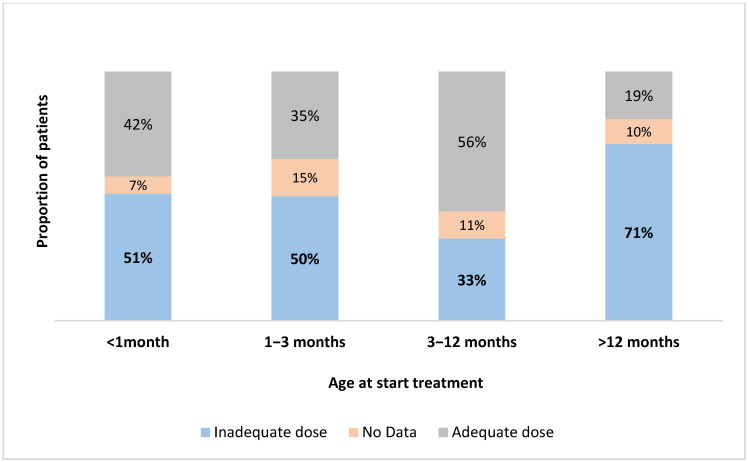
Age at LT-4 treatment at initiation.

**Figure 6 IJNS-11-00078-f006:**
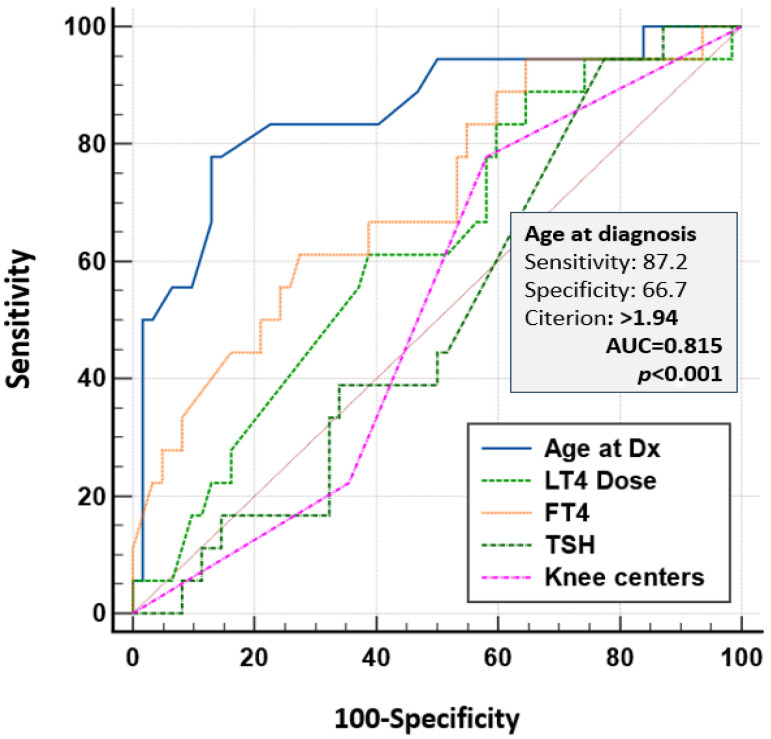
ROC comparison of risk factors for intellectual disability. ROC: Receiver Operating Characteristic (ROC); LT4: Levothyroxine.

**Figure 7 IJNS-11-00078-f007:**
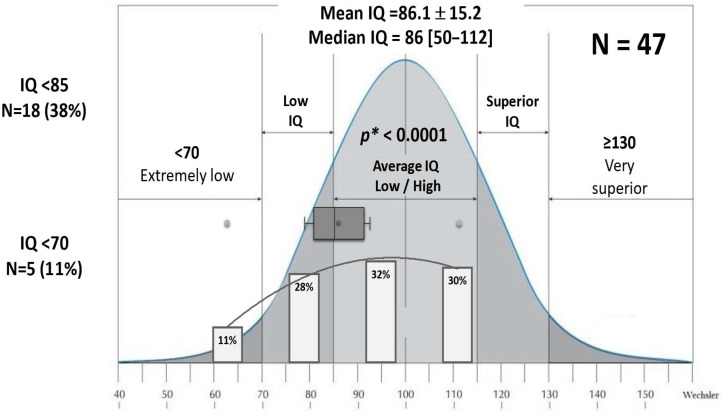
The results of the IQ assessment in 47 patients. * Comparison with the theoretical mean.

**Figure 8 IJNS-11-00078-f008:**
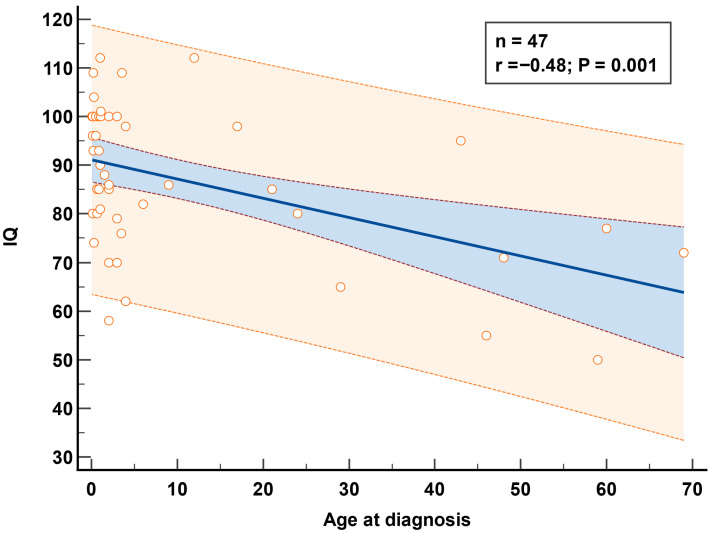
Correlation between age at diagnosis and IQ. Blue: 95% confidence interval; brown shading: 95% prediction interval.

**Table 1 IJNS-11-00078-t001:** Family history, etiological groups, and comorbidities.

Family History		
Consanguinity	*n*, (%)	75 (26%)
Family history of CH in a sibling	*n*, (%)	27 (9%)
**Etiology**		
** Dysgenesis**	*n*, (%)	109 (38%)
Athyreosis	*n*, (%)	52 (18%)
Ectopy	*n*, (%)	17 (6%)
Hypoplasia	*n*, (%)	40 (14%)
** Gland in situ**	*n*, (%)	150 (52%)
Goiter	*n*, (%)	25 (9%)
** Undetermined**	*n*, (%)	29 (10%)
**Associated abnormalities**	*n*, (%)	44 (15%)
Heart defects	*n*, (%)	23 (8%)
Renal defects	*n,* (%)	9 (3%)

**Table 2 IJNS-11-00078-t002:** Biology of CH patients at diagnosis.

Biological Data		
TSH mU/L *	Median (range)	65.35 (8.12–>100)
fT4 pmol/L	Mean ± SDS (range)	7.6 ± 6.2 (0.01–27)
TSH >100 mU/L	*n*, %	125 (43.4%)
TSH 40–100 mU/L	*n*, %	54 (18.7%)
TSH 20–40 mU/L	*n*, %	32 (11.1%)
TSH < 20 mU/L	*n*, %	77 (26.7%)
fT4 < 5 pmol/L	*n*, %	102 (35.4%)

* TSH values >100 mU/L were recorded at 100 mU/L.

**Table 3 IJNS-11-00078-t003:** A comparison of the clinical and biological data according to the etiological group.

	Global CH Cohort *N* = 288	GIS/Goiter *N* = 150	Ectopy/Hypoplasia *N* = 57	Athyreosis *N* = 52	Undetermined *N* = 29	*p*-Value
Age at diagnosis (months)	2	1.5	4	2	0.86	0.001 *
Median (range)	(0.1–375)	(0.05–51)	(0.07–155)	(0.06–150)	(0.13–32)	
M/F	129/159	80/69	24/33	10/42	15/15	0.0003
Sex ratio	0.81	1.2	0.7	0.24	1	
Consanguinity% (*N*)	26% (75)	31% (47)	23% (13)	13.5% (7)	27% (8)	0.078
TSH mU/L	65.3	50	96.3	100	47.5	0.034 *
Median (range)	(8.1–>100)	(8.1–>100)	(10.1–>100)	(17.9–>100)	(10–>100)	
TSH > 100 mU/L% (*N*)	43% (125)	35% (53)	39% (22)	73% (38)	41% (12)	<0.0001
fT_4_ (pmol/L)	7.6 ± 6.4	8.8 ± 6.1	7.2 ± 5.5	2.6 ± 2.7	8.8 ± 2.3	0.003
Mean ± SD (range)	(0.01–25.4)	(0.01–25.7)	(0.02–27)	(0.01–10.6)	(0–25.4)	
fT_4_ < 5 pmol/L %, (*N*)	35% (102)	26% (39)	37% (21)	60% (31)	38% (11)	0.0002

* Comparison between in situ gland (GIS)/dysgenesis (all data).

**Table 4 IJNS-11-00078-t004:** A comparison of clinical and biological data between permanent and transient CH in 47 patients who underwent a clinical reassessment at age 3.

	Transient CH *N* = 24	Permanent CH *N* = 23	*p*
Sex ratio M/F	12/12	8/15	0.297
Age at diagnosis in months—median (range)	4.3 (0.16–46)	2.4 (0.1–12)	0.440
TSH mU/L—median (range)	15.98 (4.3–>100)	100 (7.6–>100)	0.002
TSH > 100 mU/L, *n*, (%)	3 (12.5%)	11 (48%)	0.009
TSH < 10 mU/L, *n*, (%)	4 (17%)	0 (0%)	0.037
fT_4_ pmol/L—mean (range)	13.4 (0.1–17.2)	7.5 (0.1–18.1)	0.003
fT_4_ < 10 pmol/L *n*, (%)	5 (21%)	14 (61%)	0.011
Gland in situ *n*, (%)	23 (96%)	10 (43%)	0.001
Dysgenesis *n*, (%)	1 (5%)	13 (57%)	

**Table 5 IJNS-11-00078-t005:** Neurodevelopmental data.

Neurodevelopmental Data	*N*, (%)	Age at Evaluation Mean ± SD	Age Treatment Median	Treatment *n*, %	
				≤1 Month	≥3 Months
Psychomotor delay	43 (15%)	23.3 ± 25.1 months	13.9 months	1/43 (2%)	36/43 (84%)
Language delay	16 (6%)	18.4 ± 18.1 months	3.6 months	2/16 (12.5%)	9/16 (56%)
School-aged children	88/206 (43%)	11.6 ± 3.4 years	3 months	20/88 (23%)	47/88 (53%)
School failure	27/88 (31%)	10.8 ± 3.9 years	4 months	6/27 (22%)	16/27 (59%)
**IQ assessment *N = 47***					
Normal IQ ≥ 85	29 (62%)	5.1 ± 1.7 years	1.1 months	13/29 (45%)	9/29 (31%)
Low IQ < 85	13 (28%)	5.2 ± 1.8 years	3 months	2/13 (15%)	8/13 (61.5%)
Very low IQ < 70	5 (11%)	4.8 ± 2.5 years	29 months	0/5 (0%)	4/5 (80%)
Cognitive disharmony	7 (15%)	5.6 ± 1.3 years	2 months	2/7 (28.6%)	3/7 (43%)

**Table 6 IJNS-11-00078-t006:** Comparison of IQ groups.

		Global	IQ < 85	IQ ≥ 85	*p*
		*N* = 47	*N* = 18	*N* = 29	
Sex M/F		15/32	5/13	10/19	0.63
Age IQ test years—mean (range)		5.3 ± 1.8 (2.5–9.3)	5.5 ± 2.1 (3–9.3)	5.2 ± 1.7 (2.5–8)	0.68
Age at treatment in months—median (range)		2 (0.1–69)	3.75 (0.2–69)	1.1(0.1–42.7)	0.01
Age start L-T_4_	≤1 month (*n*,%)	16 (25.5%)	3 (17%)	13 (45%)	0.05
	>1 month (*n*, %)	31 (74.5%)	15 (83%)	16 (55%)	
Etiology	Dysgenesis	25 (53%)	13 (72%)	12 (41%)	0.05
	Gland in situ	21 (45%)	5 (28%)	16 (55%)	
L-T_4_ Dose µg/kg/day	<9 µg/kg/day	27 (54%)	12 (67%)	15 (52%)	0.26

**Table 7 IJNS-11-00078-t007:** Original studies about age at diagnosis in unscreened CH patients.

Reference (Original Study)	Country	Period	Diagnosed in Neonatal Period	Late Diagnosis Rates	Neurodevelopmental Outcome
Raiti 1971 [15]	UK	–	6% < 1st month 22% < 3 months	16% < 6 months 55% by 2 years	IQ < 90 (50%) 53% treatment > 6 months
De Jonge 1976 [20]	Netherlands	1972–1974	10% <1 month	50% at 3 months	34% IQ > 90, 17% IQ < 50
Alm 1978 [3,16]	Sweden	1969–1975	20%	52% after 3 months	41% IQ < 85 and/or neurological abnormalities
Wolter 1979 [21]	Belgium	–	7% < 1st month 46% < 3 months	21% > 1 year	IQ < 80 (23%); Normal IQ if treated <3 months
Jacobsen 1981 [22]	Denmark	1970–1975	10% 1st month	70% by 1 year	46% intellectual disability
Tarim 1992 [23]	Turkey	1964–1989	3.1%	55.4% after 2 years	21.4% inability to speak 18.1% inability to walk
Nasheiti 2005 [24]	Iraq	1993–2003	25%	75% beyond neonatal period	47.5% intellectual disability
Chen 2013 [25]	South Asian countries	1997–2008	Taiwan 56% <3 months	22% > 1 years Pakistan, India 70% >1 year	Diagnosis >3 months, higher risk of developmental delay (HR = 1.97)
Niang 2016 [26]	Senegal	2001–2014	7%	78.5% >6 months	73% intellectual disability 2.5% at school
Deliana 2016 [27]	Indonesia	1992–2002	Minimal	53% at 1–5 years 6.7% after 12 years	62.5% intellectual disability
Saoud 2019 [28]	Syria	2008–2012	>25% 1st month	75% beyond neonatal period	37.1% psychomotor delay 81% diagnosis >6 months
Kahssay 2025 [29]	Kenya *	2015–2020	5% < 1st month	80% 6–11 months 15% > 1–2 years	60% developmental delay
Our study	Algeria	2005–2023	35% < 1st month	65% ≥1 month	28% IQ < 85, 11% IQ < 70

* Facility-based study, HR: Hazard ratio.

## Data Availability

The data presented are available upon request from the corresponding author.

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
