# Peer review of "The Burden of Congenital Hypothyroidism Without Newborn Screening: Clinical and Cognitive Findings from a Multicenter Study in Algeria"

_2409-515X, 2025, doi:10.3390/ijns11030078_

Round 1

Reviewer 1 Report

Comments and Suggestions for Authors

Dear Authors,

Thank you for the opportunity to review your manuscript. The study provides valuable insight into the clinical and neurodevelopmental outcomes of congenital hypothyroidism in the absence of a national newborn screening program. The multicenter design and long-term follow-up are strengths. However, several issues require clarification or revision. Please consider the following comments:

  1. Line 85: "The exclusion criteria where" should be revised to "The exclusion criteria were".
  2. Line 102 (Section 2.5: Neurodevelopmental Assessment): Please clarify how comparisons were made across centers when different assessment tools were used.
  3. Line 162: "BMI > +2 DS" should be corrected to "BMI > +2 SD".
  4. Line 172, Table 2 and Table 3: If values >100 mU/L were excluded due to lack of an exact number, how were the upper TSH ranges (e.g., 7100) determined in Table 2 and Table 3?
  5. Table 3: The "Undetermined" group may be better placed as the final column for improved readability.
  6. Line 188: The conclusion that "The dysgenesis group showed significantly more severe forms than the in-situ gland group" should be interpreted with caution due to the exclusion of cases with TSH >100 mU/L.
  7. Line 188–190: Comparison between in situ gland and dysgenesis based on incomplete TSH data should be concerned.
  8. Figure 5: The visual presentation is unclear. Consider replacing it with a colored graph to enhance clarity and avoid confusion.
  9. Treatment Timing: Is there any available data on the time interval from diagnosis to treatment initiation? This would be highly relevant to outcome interpretation.
  10. Table 3 and Table 4: There is inconsistency in decimal separator usage. For instance, both “0,06” and “0.81” appear.
  11. Table 5: Please indicate where the "Age at evaluation" data for the school-age data group is presented.
  12. Table 5: Consider reclassifying results into either transient vs permanent CH or early vs late treatment groups to enhance interpretability.
  13. Transient CH exclusion: Were children with transient CH excluded prior to IQ analysis? Please clarify.
  14. Table 7: The table summarizing the literature review includes age at diagnosis and late diagnosis rates but omits clinical or developmental outcomes. If such outcomes are not included, the value of this table is limited. Consider revising or removing it.
  15. Lines 293–297: These lines are redundant and could be condensed for clarity.
  16. Loss to Follow-up: Please clarify whether there were any patients lost to follow-up. If so, describe how this was handled statistically, as this may significantly bias neurodevelopmental outcome interpretation.
  17. Gene Symbols: All gene symbols (e.g., DUOX2PAX8) should be italicized per standard nomenclature conventions.
  18. Classification Definitions: You classify CH as thyroid dysgenesis and dyshormonogenesis but also refer to “gland in situ”. Please clearly define how “gland in situ” fits within this classification scheme.
  19. Genetic Etiologies (Lines 326–328): The link between consanguinity and congenital anomalies should be interpreted in the different inheritance patterns of CH-related genes. For example, PAX8NKX2-1, and NKX2-5 (AD) often associated with extrathyroidal anomalies, whereas DUOX2 and TPO (AR) typically are not. Please revise this section accordingly for genetic accuracy.
  20. Initial L-T4 dosing: It is concerning that 52% of patients received an inadequate initial dose of L-T4. Please clarify the reasons for this finding and its implications.
  21. Line 386–388: While it is stated that euthyroidism was achieved in all patients at the time of neurocognitive assessment, the timing may have been too late to mitigate neurodevelopmental effects. Emphasize the importance of early thyroid normalization.
  22. Line 394: You classify cognitive impairment risk using a 2-month treatment cutoff, whereas the literature typically uses a 3-month threshold. Please explain the rationale for this deviation.
  23. Reference Formatting: Please ensure that all references are formatted according to the journal’s style guide.

I appreciate your efforts in addressing an important public health issue in a resource-limited context. I hope these comments are helpful in strengthening the manuscript.

Kind regards,

Author Response

Comments 1: Line 85: "The exclusion criteria where" should be revised to "The exclusion criteria were".

Response 1: Thank you for your careful reading. We have corrected the grammatical error as suggested "where" has been changed to "were" in the sentence referring to the exclusion criteria.

 Line 86

Comments 2: Line 102 (Section 2.5: Neurodevelopmental Assessment): Please clarify how comparisons were made across centers when different assessment tools were used.

Response 2:  Thank you for this relevant observation. Among the 47 IQ assessments performed, 37 were conducted in the same center using standardized and comparable methods. The remaining assessments were carried out in different centers but used validated psychometric tools appropriate for the child’s age. In the revised manuscript, we will clarify this point and specify that the majority of assessments (79%) were performed in a single center, which limits inter-center variability and enhances the consistency of the outcome data.

Line 111-115

Line 265-267

Comments 3: Line 162: "BMI > +2 DS" should be corrected to "BMI > +2SD".

Response 3: Thank you for pointing this out. We have corrected the terminology as suggested — "BMI > +2 DS" has been revised to "BMI > +2SD"

Line 175.

Comments 4: Line 172, Table 2 and Table 3: If values >100 mU/L were excluded due to lack of an exact number, how were the upperTSH ranges (e.g., 7100) determined in Table 2 and Table 3?

Response 4: Thank you for your observation. To clarify, TSH values reported as >100 mU/L were not excluded from the analysis. Instead, they were noted and treated as equal to 100 mU/L for the purpose of categorization and statistical analysis. This allowed us to include these values in the upper range category (71–100 mU/L) as reflected in Tables 2 and 3. This clarification will be added to the methods and table legend in the revised version of the manuscript to avoid any ambiguity.

Line 92-93

Comments 5: Table 3: The "Undetermined" group may be better placed as the final column for improved readability.

Response 5: Thank you for this helpful suggestion. We agree that placing the "Undetermined" group as the final column would improve the readability of Table 3. We will revise the table accordingly in the updated version of the manuscript.

Line 211

Comments 6: Line 188: The conclusion that "The dysgenesis group showed significantly more severe forms than the in-situ gland group"should be interpreted with caution due to the exclusion of cases with TSH >100 mU/L.

Response 6: Thank you for your comment. We would like to clarify that cases with TSH >100 mU/L were not excluded from the analysis. Although exact values above 100 mU/L were not reported, these cases were included by assigning a value of 100 mU/L for analytical purposes. This approach allowed us to retain all relevant data and accurately compare the severity between groups. We will update the manuscript to clearly state this methodology and avoid any misunderstanding regarding data inclusion.

Line 92-93

Comments 7: Line 188–190: Comparison between in situ gland and dysgenesis based on incomplete TSH data should be concerned.

Response 7: Thank you for your comment. As mentioned in our previous response, cases with TSH >100 mU/L were not excluded but recorded as 100 mU/L. Therefore, the comparison between the in-situ gland and dysgenesis groups is based on complete data. This clarification will be added to the revised manuscript.

Line 212

Comments 8: Figure 5: The visual presentation is unclear. Consider replacing it with a colored graph to enhance clarity and avoid confusion.

Response 8: Thank you for this helpful suggestion. We agree that the original version of Figure 5 could be improved for clarity. In the revised manuscript, we will replace the current version with a colored graph to facilitate interpretation. In addition, we will present the data in percentages rather than absolute numbers to make the distribution across groups more immediately understandable for readers.

Line 225

Comments 9: Treatment Timing: Is there any available data on the time interval from diagnosis to treatment initiation? This would be highly relevant to outcome interpretation.

Response 9: Thank you for this pertinent observation. Yes, data on the time interval between diagnosis and initiation of treatment were available for the majority of patients. We agree that this variable is highly relevant to interpreting neurodevelopmental outcomes. We will include this information in the revised version of the manuscript, both in the results and discussion sections. Specifically, we will provide the median and range of the time to treatment initiation and explore its association with neurocognitive outcomes where applicable.

Line 218-220

Line 393-399

Comments 10: Table 3 and Table 4: There is inconsistency in decimal separator usage. For instance, both “0,06” and “0.81” appear.

Response 10: Thank you for highlighting this inconsistency. We will revise Tables 3 and 4 to ensure uniform use of the decimal separator throughout the manuscript, in accordance with journal formatting guidelines.

Line 211, Line 233

Comments 11: Table 5: Please indicate where the "Age at evaluation" data for the school-age data group is presented.

Response 11: Thank you for your comment. The "Age at evaluation" for the school-aged data group will be added to Table 5

Line 256

Comments 12: Table 5: Consider reclassifying results into either transient vs permanent CH or early vs late treatment groups to enhance interpretability.

Response 12: Thank you for the suggestion. We agree that reclassifying the results could improve clarity. Accordingly, we will revise the presentation of data to include comparisons between early (treatment <1 month) vs. late (treatment>3months) groups, in order to enhance interpretability

Line 256

Comments 13:  Transient CH exclusion: Were children with transient CH excluded prior to IQ analysis? Please clarify.

Response 13:: Thank you for your question. Cases of transient CH without follow-up who discontinued treatment in early life (suggesting a false positive rather than transient CH) were not included in the study and were not analysed. Among our cohort, three children with transient CH underwent an IQ assessment. We will clarify this point in the revised manuscript.

Line-87

Line 263-265

Comments 14: Table 7: The table summarizing the literature review includes age at diagnosis and late diagnosis rates but omits clinical or developmental outcomes. If such outcomes are not included, the value of this table is limited. Consider revising or removing it.

Response 14:

Thank you for this relevant observation. We agree that the inclusion of clinical and neurodevelopmental outcomes would enhance the relevance and interpretability of the literature review table. Accordingly, we will revise the table to include available data on neurodevelopmental outcomes (e.g., IQ scores, cognitive delays, school performance) when reported. This addition will strengthen the comparative value of the table and better contextualize our own findings within the existing body of evidence.

Line 320

Comments 15: Lines 293–297: These lines are redundant and could be condensed for clarity.

Response 15: Thank you for your suggestion. We agree with your observation and will revise lines 293–297 to remove redundancy and improve clarity in the revised manuscript.

Line 327-331

Comments 16: Loss to Follow-up: Please clarify whether there were any patients lost to follow-up. If so, describe how this was handled statistically, as this may significantly bias neurodevelopmental outcome interpretation.

Response 16: Thank you for your important comment. Yes, a number of patients were lost to follow-up, particularly for long-term neurodevelopmental assessment. This was primarily due to the retrospective nature of the study, variability in center resources, and differences in follow-up practices. IQ data were only available for 47 out of 288 patients. We acknowledge that this limited sample may introduce a selection bias. We will revise the manuscript to clearly state this limitation in the discussion section, and highlight its potential impact on the generalizability of neurodevelopmental outcomes. No statistical imputation was applied, and the analysis was limited to available data.

Line 414-417

Comments 17: Gene Symbols: All gene symbols (e.g., DUOX2,PAX8 ) should be italicized per standard nomenclature conventions.

Response 17: Thank you for pointing this out. We will carefully review the manuscript and ensure that all gene symbols are italicized in accordance with standard nomenclature conventions.

Line 351, Line 353, Line 361-368, Line 370

Comments 18: Classification Definitions: You classify CH as thyroid dysgenesis and dyshormonogenesis but also refer to “gland in situ”. Please clearly define how “gland in situ” fits with in this classification scheme.

Response 18: Thank you for this important comment. To clarify, we used “gland in situ” refers to patients with a normally located thyroid gland. As goiter was not consistently present, and neither perchlorate discharge testing or genetic analyses were performed, a definitive diagnosis of dyshormonogenesis could not be established. We will revise the manuscript to explicitly define this classification to avoid any ambiguity.

Line 103-104

Line 354-356

Comments 19: Genetic Etiologies (Lines 326–328): The link between consanguinity and congenital anomalies should be interpreted in the different inheritance patterns of CH-related genes. For example, PAX8,NKX2-1, and NKX2-5 (AD) often associated with extrathyroidal anomalies, whereas DUOX2 and TPO (AR) typically are not. Please revise this section accordingly for genetic accuracy.

Response 19: Thank you for this important clarification. We agree that the relationship between consanguinity and congenital anomalies should be interpreted in light of the specific inheritance patterns of CH-related genes. In the revised manuscript, we will update this section to distinguish between autosomal dominant genes which are more often associated with syndromic forms or extrathyroidal anomalies, and autosomal recessive genes typically linked to isolated thyroid dysfunction. This revision will help provide a more accurate interpretation of the potential genetic mechanisms involved in our cohort.

Line 359-361

Line 364-371

Comments 20: Initial L-T4 dosing: It is concerning that 52% of patients received an inadequate initial dose of L-T4. Please clarify the reasons for this finding and its implications.

Response 20: Thank you for this valuable comment. The high proportion of suboptimal initial L-T4 dosing (52%) likely reflects multiple factors, including variability in physician practice, partial adherence to guidelines in effect at the time, and occasional hesitancy to prescribe higher doses in neonates. While detailed motivations were not systematically documented, this finding underscores the need to reinforce guideline-based practices and ensure early and adequate treatment initiation. We will expand on this point in the revised manuscript to clarify both the reasons and the clinical implications.

Line 393-398

Comments 21: Line 386–388: While it is stated that euthyroidism was achieved in all patients at the time of neurocognitive assessment, the timing may have been too late to mitigate neurodevelopmental effects. Emphasize the importance of early thyroid normalization.

Response 21: Thank you for this insightful comment. We agree that achieving euthyroidism at the time of neurocognitive assessment does not necessarily prevent early neurodevelopmental impacts. We will revise the manuscript to emphasize the critical importance of early thyroid hormone normalization—particularly during the first weeks of life—to support optimal neurodevelopmental outcomes.

Line 434-436

Comments 22: Line 394: You classify cognitive impairment risk using a 2-month treatment cutoff, whereas the literature typically uses a 3-month threshold. Please explain the rationale for this deviation.

Response 22: Thank you for this pertinent observation. The choice of a 2-month treatment cutoff in our study is based on the results of our ROC analysis presented in Figure 6, which identified 2 months as the most discriminative threshold associated with cognitive outcomes in our cohort. We will clarify this rationale in the revised manuscript to ensure alignment with your comment and to justify our deviation from the commonly cited 3-month threshold.

Line 445-446

Comments 23: Reference Formatting: Please ensure that all references are formatted according to the journal’s style guide.

Response 23: Thank you for the reminder. We have reviewed and updated all references to ensure compliance with the journal’s formatting requirements. In addition, we have added several recent and relevant references to strengthen the manuscript’s scientific context, particularly in the literature review and discussion sections

Ref 3 Line 529 (1981), Ref 35 Line 599 (2024)

Ref 50 line 622(2025), Ref 55 line 648 (2022)

Ref 57 line 652 (2023), Ref 77 line 701 (2021)

Reviewer 2 Report

Comments and Suggestions for Authors

Title: The Burden of Congenital Hypothyroidism Without Newborn Screening: Clinical and Cognitive Findings from a Multicenter Study in Algeria

This article addresses the major issue of congenital hypothyroidism (CH) in Algeria, a country where a national biochemical newborn screening (NBS) program is not yet fully implemented. CH is one of the most common preventable causes of intellectual disability in children. The lack of universal NBS leads to delayed diagnosis and treatment, which may result in irreversible neurodevelopmental damage. The primary aim was to determine the age at diagnosis of CH in Algerian children. Secondary objectives included describing the clinical and biological characteristics at presentation, identifying etiological patterns, assessing treatment adequacy, evaluating the neurodevelopmental consequences of delayed diagnosis, and analyzing the correlation between age at diagnosis and cognitive outcomes. Ultimately, the study supports the implementation of a national NBS program in Algeria.

Strength:

- Robust multicenter study: The analysis of 288 cases over nearly two decades across 20 pediatric centers offers a comprehensive overview of CH in Algeria.

- Clear identification of risk factors: The study highlights key elements affecting neurodevelopmental outcomes, such as age at diagnosis and etiology.

- Well-structured statistical analysis, including significant correlations between IQ, age at treatment initiation, and hormone levels.

- Relevant public health impact: The findings strongly support the need for a national newborn screening (NBS) program in Algeria.

Study Limitations:

- The retrospective design may introduce biases and incomplete or non-standardized data collection.

- Non-exhaustive patient recruitment limits the generalizability of the findings to the entire Algerian population.

- The limited number of IQ assessments (only 47 children) restricts the ability to draw strong conclusions about the overall cognitive impact.

- As a hospital-based study, it does not allow estimation of CH prevalence in the general population.

Specific Comments:

1) The high proportion of patients (52%) receiving suboptimal initial L-T4 dosing raises important questions about clinical practice. It would be helpful for the authors to clarify whether the low doses were due to delayed follow-up, lack of guideline adherence, resource constraints, or physician hesitancy. This explanation is crucial for understanding systemic gaps in treatment management.

2) The study mentions diagnoses up to 150 months (12.5 years), but no specific outcomes are provided for these extremely delayed cases. Given the known impact of prolonged, untreated CH, the inclusion of cognitive or clinical outcomes for this subgroup would provide valuable insight into the long-term burden and missed intervention opportunities.

3) In lines 153–159, jaundice is cited as the cause of referral in 36.5% of cases, while later it is described as being present in 58%. Please clarify that the former refers to the reason for referral, and the latter to clinical observation at diagnosis, to avoid confusion.

4) The text reports that children were diagnosed up to 150 months, but Figure 4 shows the highest diagnosis age to be under 100 months. Were the later-diagnosed patients excluded from the figure for visualization purposes? A clarification in the legend or results section would improve consistency.

5) Table 7 includes studies from the 1970s and 1980s. While this adds historical perspective, the authors should acknowledge the limitations of comparing contemporary data to older studies due to changes in health systems, diagnostics, and definitions over time.

6) Only 47 of 288 patients (16%) had IQ assessments. Were these patients selected for specific clinical reasons? The potential for selection bias should be discussed, and the generalizability of cognitive outcomes clarified.

7) The reported 51% rate of transient CH among reassessed patients is high. The authors might compare this with rates in screened populations and discuss whether iodine status, assay thresholds, or overdiagnosis could contribute.

8) In line 247, the authors report a significant negative correlation between IQ and age at diagnosis (r = –0.51, p = 0.001), indicating that earlier diagnosis is associated with higher IQ. However, in line 391, they state that the proportion of children with subnormal IQ (13.3%) was independent of age at diagnosis, L-T4 dose, and time to normalization. This appears contradictory. If there is a statistically significant correlation, it should be consistently reflected in the conclusions. The authors should clarify whether the discrepancy results from different statistical approaches.

Author Response

Comments 1: The high proportion of patients (52%) receiving sub optimal initial L-T4 dosing raises important questions about clinical practice. It would be helpful for the authors to clarify whether the low doses were due to delayed follow-up, lack of guideline adherence, resource constraints, or physician hesitancy. This explanation is crucial for understanding systemic gaps in treatment management.

Response 1: Thank you for this important comment. We acknowledge that the high proportion of patients receiving suboptimal initial L-T4 dosing warrants further explanation. Based on available clinical records, the low initial doses appear to reflect a combination of factors, including variability in physician practice, hesitancy to initiate higher doses in very young infants, and, in some cases, limited adherence to evolving guidelines at the time of diagnosis. While we do not have detailed data on each prescribing decision, we will revise the manuscript to discuss these possible contributing factors and highlight the need for greater awareness and consistency in early treatment practices.

Line 393-399

Comments 2: The study mentions diagnoses up to 150 months (12.5 years), but no specific outcomes are provided for these extremely delayed cases. Given the known impact of prolonged, untreated CH, the inclusion of cognitive or clinical outcomes for this subgroup would provide valuable insight into the long-term burden and missed intervention opportunities.

Response 2: Thank you for this highly relevant comment. We fully agree that reporting outcomes for patients diagnosed at an extremely delayed age would provide valuable insights. In response to your suggestion, we will add a specific analysis of the clinical and cognitive outcomes for this subgroup in the revised manuscript. This addition will help illustrate the long-term consequences of delayed diagnosis and highlight the importance of early detection and intervention.

Line 151

Line 246-247

Comments 3: In lines 153–159, jaundice is cited as the cause of referral in 36.5% of cases, while later it is described as being present in 58%. Please clarify that the former refers to the reason for referral, and the latter to clinical observation at diagnosis, to avoid confusion.

Response 3: Thank you for this helpful observation. You are correct, jaundice was the reason for referral in 36.5% of cases, whereas it was clinically observed at diagnosis in 58% of patients. We will revise the manuscript to clearly distinguish between the referral reason and clinical findings in order to avoid confusion.

Line 166-168

Comments 4: The text reports that children were diagnosed up to 150 months, but Figure 4 shows the highest diagnosis age to be under 100 months. Were the later-diagnosed patients excluded from the figure for visualization purposes? A clarification in the legend or results section would improve consistency.

Response 4: Thank you for your comment. You are correct, patients diagnosed after 100 months were excluded from Figure 4 for visualization purposes, as they represented a very small number of cases and skewed the scale. We will clarify this point in the figure legend.

Line 199

Comments 5: Table 7 includes studies from the 1970s and 1980s. While this adds historical perspective, the authors should acknowledge the limitations of comparing contemporary data to older studies due to changes in health systems, diagnostics, and definitions overtime.

Response 5: Thank you for this relevant comment. We acknowledge that health systems and diagnostic practices have evolved significantly over the decades. However, the studies included from the 1970s and 1980s were conducted before the implementation of neonatal screening in their respective countries, and they remain among the only available data reflecting outcomes in this populations. Our intention was not to directly compare these historical cohorts with our study, but rather to provide contextual benchmarks, given the absence of neonatal screening in our setting. To enhance the table’s relevance and interpretability, we also added neurodevelopmental outcome data where available.

Line 320

Comments 6: Only 47 of 288 patients (16%) had IQ assessments. Were these patients selected for specific clinical reasons? The potential for selection bias should be discussed, and the generalizability of cognitive outcomes clarified.

Response 6: Thank you for this important comment. The limited number of IQ assessments (47 out of 288 patients) was primarily due to the lack of availability of standardized cognitive testing in the majority of participating centers, rather than selective clinical indication. We acknowledge the potential for selection bias and will add a statement in the discussion to address this limitation and to clarify that the generalizability of cognitive outcomes may be limited.

Line 414-417

Comments 7: The reported 51% rate of transient CH among reassessed patients is high. The authors might compare this with rates in screened populations and discuss whether iodine status, assay thresholds, or overdiagnosis could contribute.

Response 7: Thank you for this relevant comment. We agree that the 51% rate of transient CH among reassessed patients appears high compared to most screened populations. Several factors may contribute to this finding, including the absence of systematic neonatal screening in our cohort, and potential iodine imbalance in the population. Overdiagnosis may also play a role, particularly in borderline cases. We will add a comment in the revised manuscript.

Line 381-386

Comments 8: In line 247, the authors report a significant negative correlation between IQ and age at diagnosis (r = –0.48, p = 0.001), indicating that earlier diagnosis is associated with higher IQ. However, in line 391, they state that the proportion of children with subnormal IQ (13.3%) was independent of age at diagnosis, L-T4 dose, and time to normalization. This appears contradictory. If there is a statistically significant correlation, it should be consistently reflected in the conclusions. The authors should clarify whether the discrepancy results from different statistical approaches.

Response 8: Thank you for pointing out this inconsistency. You are correct, the statement in line 391 is misleading, as our results do show a statistically significant association between IQ and age at diagnosis. The mention of independence from age at diagnosis was an error and is inconsistent with the earlier findings. Additionally, the sentence is redundant and does not contribute meaningfully to the discussion. We will therefore remove this sentence in the revised version of the manuscript to maintain clarity and consistency in our conclusions.

Line 442-446

Reviewer 3 Report

Comments and Suggestions for Authors

  1. The understanding of subclinical hypothyroidism is not included. If this has not been studied at all or cannot be studied, please explain why it was not included.
  2. Clarify which parts of the study used a retrospective design and which parts used a prospective design.

  3. The Wechsler Intelligence Scale IV was used to assess the psychomotor development of children with congenital hypothyroidism. It was a test that included children aged 6-16. However, it is unclear that the average age of the children included in this assessment was 5. Please clarify this.
  4. The plagiarism is 8% which means at the normal range. 
  5. The literature used in this article contains mostly data from 2010 to 2020, so it is necessary to add more data after 2020.

Author Response

Comments 1: The understanding of subclinical hypothyroidism is not included. If this has not been studied at all or cannot be studied, please explain why it was not included.

Response 1: Thank you for your observation. Subclinical hypothyroidism was not specifically analyzed in this study. Patients who discontinued treatment early were excluded from the analysis, and it is likely that a significant proportion of them had subclinical forms. However, due to the lack of consistent follow-up data confirming persistent mild TSH elevation without low fT4, it was not possible to clearly identify and analyze this subgroup. We will clarify this point in the revised manuscript and acknowledge it as a limitation in the discussion section.

Line 97-98

Comments 2: Clarify which parts of the study used a retrospective design and which parts used a prospective design.

Response 2: Thank you for your insightful comment. We confirm that the study had a mixed design. Clinical, biochemical, radiological, and treatment-related data were collected retrospectively from medical records. In contrast, neurodevelopmental assessments were conducted prospectively, as part of a standardized evaluation protocol for patients referred for follow-up. We will clarify this distinction in the “Methods” section of the revised manuscript.

Line 82

Comments 3: The Wechsler Intelligence Scale IV was used to assess the psychomotor development of children with congenital hypothyroidism. It was a test that included children aged 6-16. However, it is unclear that the average age of the children included in this assessment was 5. Please clarify this.

Response 3: Thank you for your comment. We apologize for the lack of clarity. While the Wechsler Intelligence Scale for Children-IV (WISC-IV) was used for children aged 6 to 16 years, children under 6 years of age were assessed using age-appropriate standardized tools such as the Wechsler Preschool and Primary Scale of Intelligence (WPPSI-III) and, in some cases, the Kohs Block Design Test. We will revise the manuscript to clearly state which cognitive assessment tools were used for each age group.

Line 111-116

Comments 4: The plagiarism is 8% which means at the normal range.

Response 4: Thank you for your observation. We remain committed to maintaining high standards of academic integrity.

Comments 5: The literature used in this article contains mostly data from2010 to 2020, so it is necessary to add more data after 2020.

Response 5: Thank you for this observation. We acknowledge that most of the references cited are from 2010 to 2020. However, recent studies reporting age at diagnosis and neurodevelopmental outcomes in the absence of neonatal screening have become increasingly rare, as most countries have implemented systematic screening programs. Nonetheless, we will update our literature review to include more recent and relevant articles published after 2020, particularly those that provide useful insights for comparison with our study population.

Ref 35 Line 599 (2024)

Ref 50 line 622(2025)

Ref 55 line 648 (2022)

Ref 57 line 652 (2023)

Ref 77 line 701 (2021)

Reviewer 4 Report

Comments and Suggestions for Authors

The paper impressively highlights the consequences of late treatment for congenital hypothyroidism on mental development, consequences that are rarely seen in countries with well-established screening programmes for the condition. It is also very important that the presentation covers the often overlooked proportion of genetically determined forms and the influence of consanguinity on the prevalence of hypothyroidism. The difficulties of setting up a newborn screening (NBS) programme with limited resources are clearly explained.

A few comments:

Abstract: Most parts of the abstract are written as sentences, but some are written in bullet point form. An abstract consisting entirely of complete sentences would be easier to read.

Lines 93-94  This definition should be critically revised. The values indicated appear to be the laboratory values used to include children in the study. However, they do not equate to a diagnosis of congenital hypothyroidism. The word 'congenital' should be removed from the first sentence, as it implies that the aetiology of hypothyroidism can be deduced from the laboratory values, which is not the case: A TSH value of >8 mU/l alone only justifies a suspicion of hypothyroidism (see international guidelines), not a final diagnosis. A low fT4 value indicates overt hypothyroidism, which may, however, be transient.

Lines 171-173 and Table 2: Excluding all TSH values above 100 mU/l from the presentation creates a false impression. It is recommended that the laboratory values are presented in classes.

Outcomes: It is very much appreciated that an assessment of transient forms was carried out, especially since, as discussed aptly in section 4.3, transient forms of an unknown number are often included in prevalence data. However, it is unclear whether the children included in the neuroassessment also included those with transient forms. It would be sensible in terms of content to exclude these children from the outcome presentation or at least mention the results exclusively for children who are presumably affected by permanent hypothyroidism.

Author Response

Comments 1: Abstract: Most parts of the abstract are written as sentences, but some are written in bullet point form. An abstract consisting entirely of complete sentences would be easier to read.

Response 1: Thank you for this helpful suggestion. We agree that consistency in format improves readability. We will revise the abstract to present all information in full, continuous sentences and remove the bullet points to ensure a more coherent and reader-friendly structure.

Line 34-36

Line 43-46

Comments 2: Lines 93-94 This definition should be critically revised. The values indicated appear to be the laboratory values used to include children in the study. However, they do not equate to a diagnosis of congenital hypothyroidism. The word 'congenital'should be removed from the first sentence, as it implies that the aetiology of hypothyroidism can be deduced from the laboratory values, which is not the case: A TSH value of >8 mU/l alone only justifies a suspicion of hypothyroidism (see international guidelines), not a final diagnosis. A low fT4 value indicates overt hypothyroidism, which may, however, be transient.

Response 2: Thank you for this important and well-founded comment. We fully agree with your observation. The laboratory values mentioned (TSH >8 mU/L and low fT4) were indeed the inclusion criteria for our study, and do not constitute, on their own, a definitive diagnosis of congenital hypothyroidism. We will revise the text to reflect this distinction more accurately and remove the term "congenital" from the initial definition, in accordance with international guidelines.

Line 96-97

Comments 3: Lines 171-173 and Table 2: Excluding all TSH values above 100mU/l from the presentation creates a false impression. It is recommended that the laboratory values are presented in classes.

Response 3: Thank you for this pertinent remark. We would like to clarify that TSH values above 100 mU/L were not excluded from the analysis. However, due to limitations in some laboratory reports, values >100 mU/L were often not provided with an exact number and were therefore recorded as "100" in the dataset. We agree with your suggestion and will revise Table 2 to present TSH values in clinically relevant classes (e.g., <20, 20–40, 40–100, >100 mU/L) to improve clarity and avoid any misleading interpretation. This clarification will also be added to the methods and table legend.

Line 92-93

Comments 4: Outcomes: It is very much appreciated that an assessment of transient forms was carried out, especially since, as discussed aptly in section 4.3, transient forms of an unknown number are often included in prevalence data. However, it is unclear whether the children included in the neuro assessment also included those with transient forms. It would be sensible in terms of content to exclude these children from the outcome presentation or at least mention the results exclusively for children who are presumably affected by permanent hypothyroidism.

Response 4: Thank you for your insightful comment. Among the children who underwent neurocognitive assessment, only 3 were later identified as having transient congenital hypothyroidism. We agree that including these few cases could introduce bias in interpreting outcomes related to permanent hypothyroidism. Therefore, in the revised version of the manuscript, we will specify this detail and, where appropriate, provide the neurocognitive results of these transient cases.

Line 263-265

Round 2

Reviewer 1 Report

Comments and Suggestions for Authors

Thank you for your thorough and complete explanation. All questions are well answered and clearly addressed.

In addition to the supplementary materials, I would like to highlight the following points:

  1. Table S1: The female-to-male ratio reported appears to be incorrect. Kindly review and clarify this discrepancy.

  2. Table S3: Please use English terminology throughout the article. Specifically, replace “moy ± SD” with “mean ± SD” and “µg/kg/j” with “µg/kg/day”.

  3. Table S4: “Ombilical hernia” should be corrected to “Umbilical hernia”.

    Best wish,

Author Response

Comment 1:  Table S1: The female-to-male ratio reported appears to be incorrect. Kindly review and clarify this discrepancy.

Response 1: Thank you for your comment. The reported value corresponded to the male-to-female ratio, which led to the misunderstanding. We have corrected the labeling to accurately reflect the female-to-male ratio.

We also realized that the version of the tables initially submitted was not the final one. We sincerely apologize for this oversight. The updated tables now include the corrected data, including supplement use.

Comment 2: Table S3: Please use English terminology throughout the article. Specifically, replace “moy ± SD” with “mean ± SD” and “µg/kg/j” with “µg/kg/day”.

Response 2: Thank you for your observation. We have replaced all non-English terms throughout the manuscript. Specifically, “moy ± SD” has been corrected to “mean ± SD” and “µg/kg/j” to “µg/kg/day” in accordance with English scientific writing conventions

Comment 3: Table S4: “Ombilical hernia” should be corrected to “Umbilical hernia”.

Response 3: Thank you for pointing this out. We have corrected the spelling from “Ombilical hernia” to “Umbilical hernia” in the manuscript.
